**Data Availability Statement:** All relevant data are within the manuscript and its Supporting Information files.

# Remdesivir in combination with dexamethasone for patients hospitalized with COVID-19: A retrospective multicenter study

Simon B. Gressens[1]◉*, Violaine Esnault[1]◉, Nathalie De Castro[1], Pierre Sellier[1], Damien Sene[2], Louise Chantelot[3], Baptiste Hervier[4], Constance Delaugerre[5], Sylvie Chevret[6], Jean-Michel Molina[1], Saint-Louis CORE group[¶]

1 Département des Maladies Infectieuses, Hôpitaux Saint Louis-Lariboisière, Université de Paris, AP-HP, Paris, France, 2 Département de Médecine Interne, AP-HP Hôpital Lariboisière, Paris, France, 3 Service de Pneumologie, AP-HP Hôpital Saint Louis, Paris, France, 4 UF de Médecine Interne, AP-HP Hôpital Saint Louis, Paris, France, 5 Laboratoire de Virologie, AP-HP Hôpital Saint Louis, Paris, France, 6 Service de Biostatistique et Information Médicale, AP-HP Hôpital Saint Louis, Paris, France

◉ These authors contributed equally to this work.
¶ Membership of the Saint Louis CORE group is listed in the Acknowledgments.
* simon.gressens@aphp.fr

## Abstract

### Background

Dexamethasone is standard of care for the treatment of patients with COVID-19 requiring oxygen. The objective is to assess the clinical benefit of adding remdesivir to dexamethasone.

### Patients and methods

A retrospective cohort study of hospitalized patients with COVID-19 pneumonia requesting low-flow oxygen who received dexamethasone. Patients admitted to infectious diseases wards also received remdesivir. Primary outcome was duration of hospitalization after oxygen initiation. Secondary outcomes were in-hospital death, and death and/or transfer to the intensive care unit. To handle potential confounding by indication bias, outcome comparison was performed on propensity score-matched populations. Propensity score was estimated by a multivariable logistic model including prognostic covariates; then 1:1 matching was performed without replacement, using the nearest neighbor algorithm with a caliper of 0.10 fold the standard deviation of the propensity score as the maximal distance. Balance after matching was checked on standardized mean differences.

### Results

From August 15th 2020, to February 28th, 2021, 325 patients were included, 101 of whom received remdesivir. At admission median time from symptoms onset was 7 days, median age: 68 years, male sex; 61%, >1 comorbidity: 58.5%. Overall 180 patients matched on propensity score were analyzed, 90 each received remdesivir plus dexamethasone or dexamethasone alone. Median duration of hospitalization was 9 (IQR: 7–13) and 9 (IQR: 5–18)

**Funding:** The author(s) received no specific funding for this work.

**Competing interests:** Jean-Michel Molina has participated to GILEAD Advisory Boards and has received research grants support for his institution in HIV research. This does not alter our adherence to PLOS ONE policies on sharing data and materials.

days with and without remdesivir, respectively (p = 0.37). In-hospital death rates and rates of transfer to the intensive care unit or death were 8.9 and 17.8% (HR: 0.46, 95% CI: 0.21–1.02, p = 0.06) and 20.0 and 35.6% with and without remdesivir, respectively (HR: 0.45, 95% CI: 0.23–0.89, p = 0.015).

## Conclusion

In hospitalized patients with COVID-19 pneumonia receiving low-flow oxygen and dexamethasone, the addition of remdesivir was not associated with shorter hospitalization or lower in-hospital mortality but may have reduced the combined outcome of death and transfer to the intensive care unit.

## Introduction

Since December 2019, the COVID-19 pandemic has rapidly spread and overwhelmed healthcare systems worldwide with more than 260 million global cases and more than 5.2 million deaths as of December 4th, 2021 [1, 2]. The disease ranges from asymptomatic cases to acute respiratory distress syndrome and death. Numerous pharmacologic agents have been studied to improve clinical outcome, and medical practice has evolved with results from clinical trials. Dexamethasone is now standard of care for the treatment of hospitalized patients with COVID-19 pneumonia requiring oxygen following the release of the results from the RECOVERY trial in July 2020 which showed a significant reduction in mortality, presumably through the control of the deleterious inflammatory response to SARS-COV-2 infection in severe patients [3, 4].

Other treatments aiming at inhibiting viral replication have been investigated. Among them, remdesivir, a nucleotide analogue prodrug with *in vitro* activity against a broad array of RNA viruses including SARS-CoV-2 has been studied in multiple clinical trials and cohort studies [5]. Initial randomized placebo-controlled trials have shown a faster time to recovery with remdesivir, but no survival benefit was demonstrated [6, 7]. Subsequent open-label randomized trials and cohort studies yielded mixed results. The two large randomized SOLIDARITY and DISCOVERY trials could not show a clinical benefit with remdesivir, and there was no evidence of reduced mortality [8, 9]. A meta-analysis not including data from the DISCOVERY trial showed a non-significant trend toward a reduction of mortality in the subgroup of non-ventilated patients receiving remdesivir and low or high-flow oxygen (RR: 0.83, 95%CI: 0.63;1.01) [8]. Large cohort studies have also reported improved clinical outcomes with remdesivir [10, 11]. In the ACTT-1 trial, the benefit of remdesivir was also most apparent in patients receiving low-flow oxygen [6]. The use of remdesivir still remains controversial especially among moderately severe patients on low-flow oxygen hospitalized in medical wards [12]. NIH guidelines recommend remdesivir for patients requiring low-flow oxygen, alone or in combination with dexamethasone while WHO issued a conditional recommendation against remdesivir in hospitalized patients with COVID-19 [13, 14]. Other guidelines, such as French guidelines, do not recommend the use of remdesivir, in particular following the release in January 2021 of the results of the DISCOVERY trial [9, 15].

However there has been limited data and no randomized trial assessing the combination of dexamethasone and remdesivir in hospitalized patients with COVID-19 pneumonia requiring low-flow oxygen [11].

We wished to assess the potential benefit on clinical outcome of the combination of remdesivir plus dexamethasone compared to dexamethasone among hospitalized patients with COVID-19 requiring low-flow oxygen and performed a retrospective analysis of patients admitted to our institution between August, 15th 2020 and February 28th, 2021.

## Patients and methods

### Study design and patient selection

We conducted a retrospective cohort study assessing in-hospital clinical outcomes among adult patients admitted for COVID-19 in five medical wards (infectious diseases (n = 2), internal medicine (n = 2) and pneumology (n = 1)) of the Saint-Louis and Lariboisière Hospitals in Paris, France, between August 15th, 2020 and February 28th, 2021. This study was approved by the ethic research committee of the French infectious disease society (IRB00011642; COVID-2021-02) which considered the study as minimal risk and patients or their relatives were informed that their data would be used anonymously for this analysis. This study was conducted during the second wave of the pandemic in France when the circulation of the SARS CoV-2 alpha variant was still limited to < 15% of the isolates in the Paris region.

During the study period, dexamethasone (6 mg/d for 10 days) was standard of care in all wards among patients with COVID-19 pneumonia requiring oxygen following the results of the Recovery trial. The combination of remdesivir to dexamethasone was only used in the two infectious diseases wards of the Saint-Louis and Lariboisière hospitals. Supplemental oxygen (< 15 L/min) was given to maintain SpO2 ≥94% during hospitalization using a mask or nasal prongs. High-flow oxygen (> 15l/min) and non-invasive ventilation were not used in the medical wards and required transfer to the ICU. Remdesivir was given at the same time as dexamethasone, as a 5-day intravenous infusion of 200 mg the first day and 100 mg/d the following days. Remdesivir was not used in patients with a creatinine clearance of < 50 ml/min or with an alanine or aspartate aminotransferase level of more than 5 times the upper limit of normal as recommended. This study was conducted before the favorable results of the Recovery trial with tocilizumab were available (online publication February 25, 2021) and tocilizumab was only used in two patients who did not receive remdesivir during the study period [16].

Patients directly admitted to the intensive care unit (ICU) or directly transferred from the emergency room to the ICU were not included in the study since we wished to assess the impact of remdesivir and dexamethasone on clinical outcome including transfer to the ICU.

All adult patients identified through the pharmacy lists as having received dexamethasone or remdesivir during the study period were assessed through their electronic health records (EHR) and included in the study if they had a confirmed SARS-CoV-2 infection defined by positive nasopharyngeal polymerase chain reaction (PCR) test with pulmonary infiltrates on chest X-ray of pulmonary CT-scanner. Exclusion criteria included direct admission to the ICU (or from the emergency room to the ICU), no oxygen-therapy during hospitalization or no corticosteroid prescription.

### Data collection

Clinical and laboratory data from the infectious diseases wards and from the internal medicine ward of the Saint-Louis hospital were retrospectively collected from patients EHR by two trained physicians (SG and VE). Data from patients admitted to the Pneumology ward of the Saint Louis Hospital and the Internal Medicine ward of the Lariboisière Hospital were collected prospectively by two other physicians (LC and DS). When remdesivir was prematurely discontinued, the patient EHR was analyzed to identify the reason for discontinuation. Severity at admission was evaluated using the WHO clinical progression scale, ranging from 0

(uninfected) to 10 (dead): moderate disease scores were 4 (hospitalized but no oxygen therapy) or 5 (hospitalized and oxygen delivered by nasal prongs or facial mask) [17].

## Outcomes

The primary outcome was the length of hospital stay after oxygen initiation. Secondary outcomes wished to assess disease progression to severe forms of the disease and included in-hospital death, in-hospital death and/or transfer to the ICU and in-hospital death and/or mechanical ventilation (MV). Safety of remdesivir was also assessed by the analysis of EHR collecting drug-related adverse events and premature treatment discontinuations.

## Statistical analysis

Categorical variables were expressed as percentages, and quantitative variables as median [interquartile range]. In-hospital mortality and other outcome measures of interest (transfer to the ICU or MV) were measured from the day of oxygen therapy initiation. Comparison of baseline characteristics between groups were based on Wilcoxon rank sum tests or exact Fisher tests, and a multivariate logistic model was used to identify predictive factors of outcomes (including age, gender, BMI, comorbidities, WHO severity score at admission and any confounding factor identified by univariate analyses). In order to account for indication bias resulting from potential differences between the two treatment groups (dexamethasone alone or dexamethasone with remdesivir), a propensity score matching analysis was performed with the main analysis to assess primary and secondary outcomes conducted on the matched populations [18, 19]. The propensity score was estimated from a multivariable logistic regression, including age, sex, obesity (BMI > 30 kg/m$^2$), diabetes, hypertension, cardiovascular comorbidity, chronic kidney disease, hematological or solid cancer, solid organ or stem cell transplantation, time from onset of symptoms to oxygen initiation, initial oxygen flow, and CRP level at admission. To each patient in the remdesivir plus dexamethasone group, one patient in the dexamethasone group was matched on the basis of the closeness of their propensity score, using the nearest neighbor algorithm. Matching was performed without replacement (once chosen, the patient in the dexamethasone group could not be chosen thereafter), using a caliper of 0.10 fold the standard deviation of the propensity score as the maximal tolerance criterion [20, 21]. The performance of the score to balance treatment groups was assessed using standardized mean differences (SMD) computed before and after matching. Estimation of treatment effect was based on the matched cohort of 180 patients (90 remdesivir plus dexamethasone and 90 dexamethasone alone), measured on the hazard of events, using Cox model with robust variance estimate to handle the matching, as well as on the prevalence of events using generalized linear models. We then tested treatment-by-subset interaction using the Gail and Simon statistics [22], considering 2 groups of time of treatment initiation (within 5 days, > 5 days from disease onset). All analyses were performed on R version 4.0.3 statistical software using the R survival package [23].

## Results

### Population

Between August 15$^{th}$ 2020 and February 28$^{th}$ 2021, 386 patients were identified through electronic queries from the pharmacy. Remdesivir was requested for 118 patients and we included 101 patients in the final analysis for the remdesivir + dexamethasone group (R+DXM), excluding 11 patients who did not receive dexamethasone, 5 directly admitted to the ICU from the emergency room, and 1 who did not receive oxygen. Remdesivir was only prescribed to patients

admitted to the infectious disease wards: 95.6% (87/91) of those admitted at the Saint-Louis hospital and 18.7% (14/75) of those admitted at the Lariboisière hospital). Dexamethasone without remdesivir was requested for 343 patients, and we excluded 119 patients, 53 patients without a confirmed COVID-19 diagnosis, 28 who did not receive dexamethasone, 28 directly admitted to the ICU from the emergency room, and 10 who did not receive oxygen. Overall, 224 patients receiving dexamethasone without remdesivir (DXM group) were included in the analysis.

Patient characteristics at the time of admission are shown in Table 1. Overall, median age was 68 (IQR 58–80) years, male sex 61%, median BMI 27 (IQR 23.7–31) kg/m$^2$. Approximately 58.5% of patients had at least two comorbidities including hypertension (53%), diabetes mellitus (33%) and cardiovascular disease (20.5%). Solid or hematological malignancies were significantly more frequent in the R+DXM than in the DXM group (23.7% vs 9.4%, respectively,

**Table 1. Baseline demographic and clinical characteristics of the patients (unmatched and matched cohorts).**

| | Unmatched cohort | | | Matched cohort | | |
|---|---|---|---|---|---|---|
| | DXM | R+DXM | p-value | DXM | R+DXM | p-value |
| | (n = 224) | (n = 101) | | n = 90 | n = 90 | |
| Age (years) | 68.6 [58.8; 79.5] | 67.7 [58.7; 77.0] | 0.69 | 67.0 [58.8; 77.9] | 68.6 [58.8; 77.9] | 0.72 |
| Female (%) | 85 (38%) | 43 (43%) | 0.46 | 35 (39%) | 37 (41%) | 0.88 |
| Ethnicity | | | 0.46 | | | 0.94 |
| White | 163 (73%) | 65 (64%) | | 62 (69%) | 58 (64%) | |
| Black | 17 (8%) | 8 (8%) | | 7 (8%) | 8 (9%) | |
| Asian | 4 (2%) | 4 (4%) | | 2 (2%) | 4 (4%) | |
| Other | 40 (17%) | 24 (24%) | | 19 (21%) | 20 (22%) | |
| BMI (kg/m$^2$) | 27.8 [23.7; 31.2] | 26.9 [24.0; 31.2] | 0.77 | 27.8 [24.8; 32.4] | 27.0 [24.2; 31.2] | 0.30 |
| Obesity (BMI > 30 kg/m$^2$) | 50 (22%) | 28 (27%) | 1.00 | 25 (28%) | 25 (28%) | 1.00 |
| Hypertension | 119 (53%) | 54 (53%) | 1.00 | 45 (50%) | 46 (51%) | 1.00 |
| Diabetes | 75 (33%) | 33 (33%) | 1.00 | 15 (28%) | 28 (31%) | 0.74 |
| Cardiovascular disease | 40 (20%) | 21 (21%) | 1.00 | 21 (24%) | 16 (18%) | 0.36 |
| Pulmonary disease | 36 (16%) | 11 (11%) | 0.24 | 11 (12%) | 11 (12%) | 1.00 |
| Tobacco usage | 30 (19%) | 7 (7%) | 0.016 | 14 (15%) | 7 (8%) | 0.16 |
| Chronic renal failure | 35 (16%) | 11 (11%) | 0.30 | 10 (11%) | 10 (11%) | 1.00 |
| Neurological disease | 38 (17%) | 17 (17%) | 1.00 | 10 (11%) | 15 (17%) | 0.39 |
| Hemopathy or solid cancer | 21 (9%) | 24 (24%) | 0.0009 | 13 (14%) | 13 (14%) | 1.00 |
| Solid organ transplant or hematopoietic stem cell transplant | 9 (4.84%) | 3 (2.97%) | 0.55 | 6 (6%) | 3 (3%) | 0.50 |
| Number of comorbidities | | | 0.16 | | | 0.44 |
| 0 | 36 (16%) | 9 (9%) | | 11 (12%) | 9 (10%) | |
| 1 | 57 (26%) | 32 (32%) | | 22 (24%) | 30 (33%) | |
| ≥2 | 130 (58%) | 60 (59%) | | 57 (63%) | 51 (57%) | |
| WHO scale score at admission | | | <0.0001 | | | <0.0001 |
| 4: hospitalized but no oxygen therapy | 73 (33%) | 10 (10%) | | 31 (34%) | 8 (9%) | |
| 5: hospitalized with oxygen by nasal prongs or mask | 151 (67%) | 91 (90%) | | 59 (66%) | 82 (91%) | |
| Time from symptoms to initiation of oxygen-therapy (days) | 7 [4; 10] | 7 [4; 10] | 0.88 | 7 [4;10] | 7 [4; 10] | 0.92 |
| Initial oxygen requirement (for SpO2 >94%) (L/min) | 2.25 [2; 4] | 2 [2; 3] | 0.01 | 2 [1; 3] | 2 [2; 3] | 0.23 |
| Laboratory tests at admission | | | | | | |
| Lymphocytes (G/L) | 0.94 [0.63; 1.25] | 0.86 [0.6; 1.15] | 0.14 | 1.00 [0.66; 1.29] | 0.91 [0.64; 1.16] | 0.20 |
| CRP (mg/L) | 40 [2; 87] | 32 [2; 66.5] | 0.11 | 37 [2; 78.8] | 34.5 [2; 68] | 0.90 |
| D-dimers (ng/mL) | 1055 [648.2; 1672] | 850 [495.5; 1575] | 0.058 | 1030 [672; 1539] | 850 [502; 1575] | 0.18 |
| Serum creatinine (mg/dL) | 0.97 [0.80; 1.22] | 0.85 [0.73; 1.11] | 0.025 | 0.83 [0.66; 0.98] | 0.75 [0.64; 0.98] | 0.39 |

p = 0.0009) whereas active smoking was significantly less frequent in the R+DXM than in the DXM group (7.3% vs 18.5%, p = 0.016).

Severity at admission evaluated through the WHO clinical progression scale, was greater in the R+DXM group than in the DXM group with a higher proportion of patients with a WHO score of 5 (90% vs. 67%, respectively, p<0.0001). Median time from onset of symptoms to oxygen initiation was similar in the two groups (7 days (IQR 4–10)) and the median time from admission to oxygen initiation was 0 day (IQR: 0–0).

There was no difference between the two groups regarding baseline lymphocyte count, CRP or D-dimers levels at admission. Baseline serum creatinine was slightly but significantly higher in the DXM group than in the R+DXM group (0.97 mg/dL [0.80;1.22] versus 0.85 mg/dL [0.73;1.11] respectively, p = 0.025). In the multivariable analysis, mortality was associated with older age (p<0.001), cardiovascular disease (p< 0.01), receipt of an organ or stem cell transplant (p< 0.001) and higher initial oxygen flow (p<0.001).

Using propensity score, 90 of the 101 patients receiving remdesivir (89%) were matched on PS to 90 control patients who only received dexamethasone. Baseline characteristics of these two matched groups are shown in Table 1. The median length of hospital stay from oxygen initiation was not significantly different between DXM and R+DXM groups: 9 days vs 9 days respectively; p = 0.37) (Table 2). Overall, 24 patients died, 8 in the R+DXM group (8.9%) versus 16 (17.8%) in the DXM group (HR: 0.47, 95% CI: 0.21–1.04, p = 0.06), and 50 patients died or were admitted to the ICU because of clinical worsening: 18 in the R+DXM group (20%) versus 32 (35.56%) in the DXM group (HR: 0.43, 95% CI: 0.23–0.81, p = 0.008). Similarly, the rate of in-hospital death or MV was 11.11% and 25.56% in the R+DXM and DXM groups, respectively (OR: 0.86, 956% CI: 0.77–0.97, p = 0.012) (Fig 1). Results were not markedly modified when further adjusting on the residual imbalance in the OMS scale at study entry (Table 2). Moreover, the forest-plot in Fig 2 showed a positive quantitative interaction (p = 0.024) between early initiation of remdesivir (within the first 5 days of symptoms onset) and treatment effect on disease progression (death or mechanical ventilation).

Finally, we did not observe any severe adverse event related to the use of the combination of remdesivir and dexamethasone in our study. Remdesivir was prematurely stopped in 5/

**Table 2. Outcomes according to treatment group in the propensity score matched populations.**

|  | Dexamethasone only (n = 90) | Remdesivir+Dexamethasone (n = 90) | Effect measure (95%CI); p-value | |
|---|---|---|---|---|
|  |  |  | Unadjusted | Adjusted on WHO scale at baseline |
|  |  |  | Mean difference | |
| Mean Length of stay (days) | 9 [5; 18] | 9 [7; 13] | 1.9 [-2.3; 6.1]; P = 0.37 | 1.4 [-3.5; 6.3]; P = 0.58 |
|  |  |  | Hazard Ratio (HR) | |
| In-hospital Death rate (n, %) | 16 (18%) | 8 (9%) | 0.47 [0.21; 1.04]; P = 0.06 | 0.46 [0.21; 1.03]; P = 0.059 |
| In-hospital Death rate or ICU admission (n, %) | 32 (35.56%) | 18 (20%) | HR: 0.43 [0.23; 0.81]; P = 0.008 | 0.42 [0.23; 0.80]; P = 0.008 |
|  |  |  | Odds Ratio (HR) | |
| In-hospital Death rate or MV (n,%) | 23 (26%) | 10 (11%) | 0.86 [0.77; 0.97]; P = 0.012 | 0.87 [0.77; 0.97]; P = 0.019 |

DXM: Dexamethasone. R+DXM: Remdesivir plus dexamethasone. MV: Mechanical ventilation. ICU: Intensive care unit.

*Either mean difference for duration of hospitalization or hazard ratio for the censored outcomes. DXM: Dexamethasone. R+DXM: Remdesivir plus dexamethasone. MV: Mechanical ventilation. ICU: Intensive care unit.

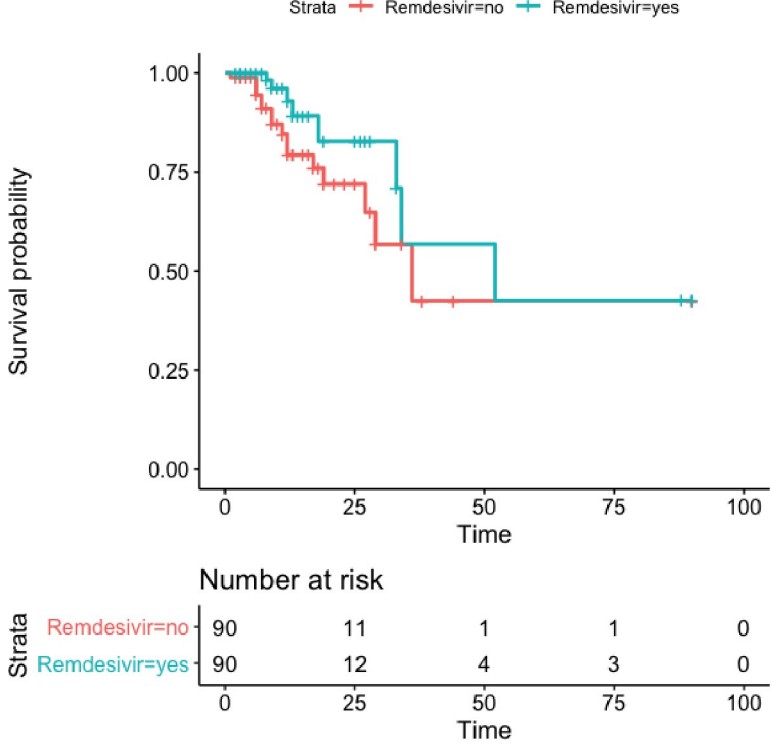

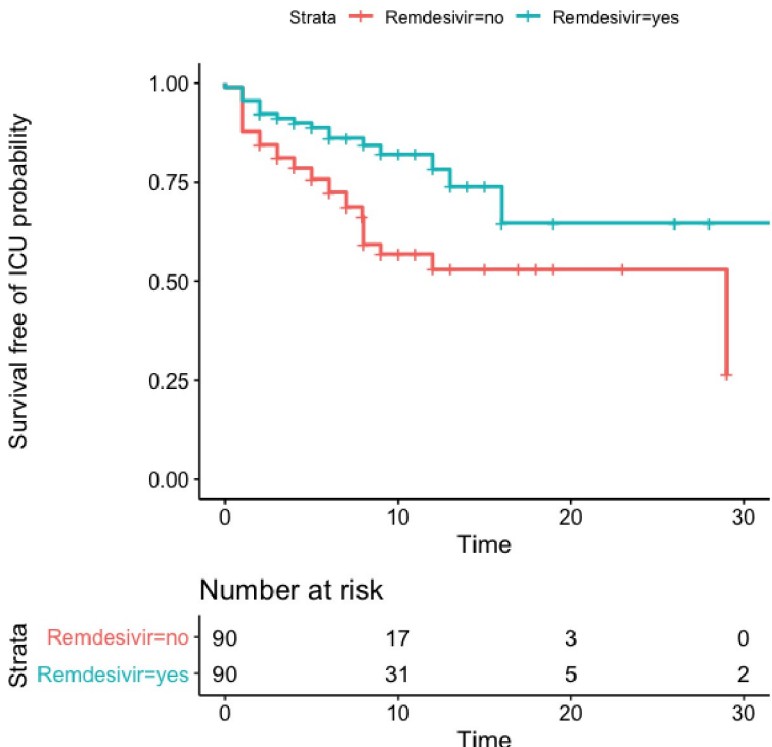

**Fig 1. Estimated outcomes (survival in upper plot, survival free of transfer to the ICU in lower plot) in the matched cohort.**

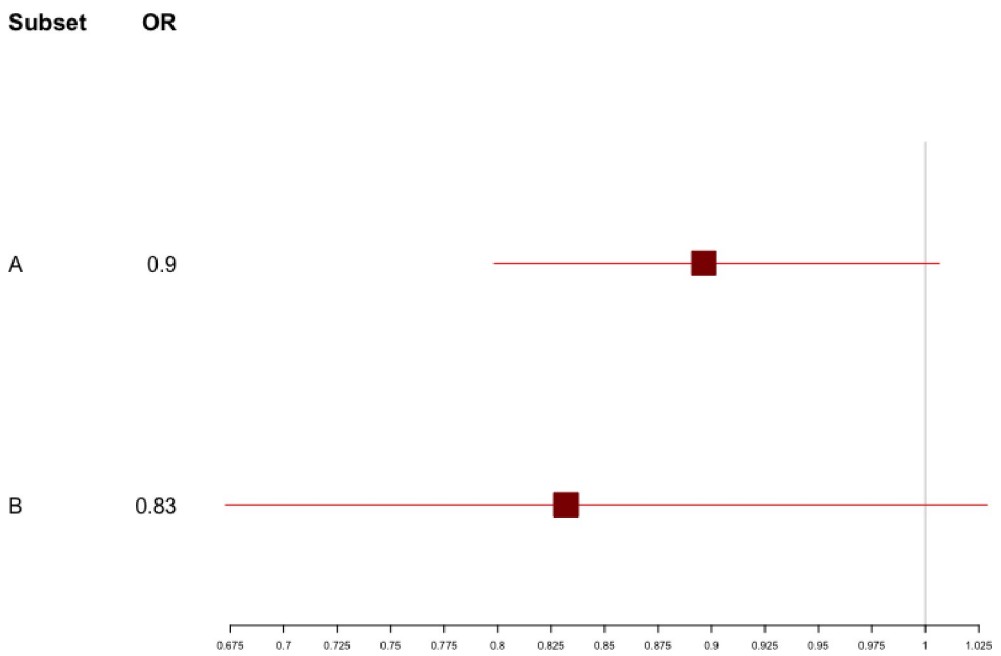

**Fig 2.** Forest plot HR of rate of death and/or MV between DXM and R+DMX groups in the matched population according to duration of symptoms before treatment initiation: A) ≥5 days, B) ≤ 5 days.

101 patients (5%) patients due to transient and asymptomatic elevation of liver enzymes. All enzyme elevation rapidly returned to normal within days after drug discontinuation.

## Discussion

In this study, we wished to assess the potential clinical benefit of adding remdesivir to dexamethasone in adult patients with moderately severe COVID-19 pneumonia hospitalized in medical wards and requiring low-flow oxygen therapy.

Indeed, whereas dexamethasone is standard of care for patients with COVID-19 pneumonia requiring oxygen supplementation, the benefit of adding remdesivir remains largely unknown.

This issue has not been addressed in randomized trials yet: in the Recovery trial, only 3 of the 2104 patients received remdesivir in the dexamethasone group, in the ACTT-1 trial only 23% received corticosteroids and remdesivir and in the SOLIDARITY trial 47.7% of patients in the remdesivir arm received corticosteroids [8].

In our study, we compared clinical outcomes in two groups of patients: those who received only dexamethasone at the time of oxygen initiation (DXM group) and those who received remdesivir in combination with dexamethasone (R+DXM group). Indeed, during the study period, 70.7% (87/123) and 18.7% (14/75) of patients admitted to the infectious disease wards of the Saint-Louis and Lariboisière hospitals, respectively, received remdesivir in combination with dexamethasone. To account for potential prescription bias, we used propensity scores and were able to match 90 (89%) of patients who received remdesivir plus dexamethasone to 90 patients who received only dexamethasone. In our propensity score-matched cohort, the association of remdesivir and dexamethasone did not reduce the length of hospital stay as we had hypothesized. This finding seems contradictory to prior reports showing in placebo-controlled studies a faster time to recovery with remdesivir but are in agreement with the results

of large open-label trials. Also, a recent meta-analysis of remdesivir trials found no effect of remdesivir on length of hospital stay [24].

Of interest secondary clinical outcomes in this PS matched populations showed a reduced in-hospital death in the R+DMX group as compared to the DXM group, though non statistically significant (HR: 0.47, p = 0.06) and a significantly lowered clinical progression to in-hospital death or transfer to the ICU or to in-hospital death and MV (HR: 0.43 and OR: 0.86, p = 0.008 and 0.012, respectively) in the R+DXM as compared to the DXM group.

These findings are contradictory to those of the Recovery trial where no benefit was found in the subgroup of patients receiving remdesivir and dexamethasone [3]. Similarly, in the DIS-COVERY trial where most patients also received corticosteroids with remdesivir, no clinical benefit was seen [9]. However, both trials aimed at assessing the clinical benefit of remdesivir and not the combination, and unlike in our study both drugs were not administered simultaneously. Our results are in agreement with those of a nationwide population-base study in Denmark reporting reduced mortality rate and need for mechanical ventilation in patients receiving remdesivir and dexamethasone as compared to standard of care, but patients in the standard of care arm did not receive dexamethasone and were not enrolled at the same time [25]. Another large retrospective cohort study in the United States reported a shorter time to clinical improvement with remdesivir as compared to matched controls but the combination of remdesivir and dexamethasone was not associated with reduced mortality compared to remdesivir alone [11]. There is therefore a need to conduct randomized trials assessing the potential clinical benefit of combining remdesivir to dexamethasone in patients with COVID-19 pneumonia requiring low-flow oxygen.

In our study, we also found a significant interaction between treatment effect and the time from symptoms onset to treatment initiation with improved effect on severe clinical outcome in patients with treatment onset within 5 days, which is consistent with previous reports showing a larger benefit of remdesivir when given early in the disease [6, 7, 26]. This is in agreement with the recent results of the Pinetree study demonstrating that a 3-day course of remdesivir was highly effective at preventing COVID-19 related hospitalization or death in high-risk non-hospitalized COVID-19 patients [27]. This finding is also consistent with the mode of action of remdesivir, which may be beneficial in the early phase of the disease with high level of viral replication, since no survival benefit was seen in patients admitted to the ICU or requiring high-flow oxygen [6, 8].

In addition, we did not observe any deleterious effect of remdesivir in our study and the safety profile of a 5-day course of remdesivir was favorable with only 5% of patients prematurely discontinuing remdesivir because of liver enzyme elevations.

Our study has however several limitations. First, it is a retrospective, observational cohort study and despite the use of propensity score to match populations, we cannot exclude residual and unmeasured confounders that may have biased our estimates of treatment effect. Second, the number of patients included in our study remains low, especially when the analysis was restricted to the matched groups, and the validity of our results needs to be confirmed in larger studies.

In conclusion, our study showed that in mild to moderately severe patients hospitalized for COVID-19 pneumonia and requiring oxygen the association of remdesivir and dexamethasone did not reduce the length of admission or the rate of in-hospital death but may have reduced the progression to severe disease defined as in-hospital death or transfer to the ICU. Clinical trials dedicated to this specific population are needed to confirm our results and assess whether patient prognosis can be improved with early administration of remdesivir in patients initiating dexamethasone.

## Supporting information

**S1 File.**
(XLSX)

## Acknowledgments

The Saint Louis CORE group is a collaborating group of clinicians, radiologists, biologists, pharmacists and clinical research assistants of Saint Louis Hospital. They all have participated to the care of patients with COVID19 and/or to research into COVID19 in Saint Louis Hospital, Paris, during the SARS-COV2 epidemic. They decided to share their data to ease local research into COVID19. All the manuscript written on behalf of the Saint Louis CORE group has been, preliminary to submission, sent to all members for critical rereading and consent for publication.

Members of the Saint Louis CORE group: Achili Y, Ades L, Aguinaga L, Archer G, Benattia A, Bercot B, Bertinchamp R, Bouaziz JD, Bouda D, Boutboul D, Brindel Berthon I, Bugnet E, Caillat Zucman S, Celli Lebras K, Chabert J, Chaix ML, Clément M, Davoine C, De Kerviler E, De Margerie-Mellon C, Delaugerre C, Depret F, Djaghout L Dupin C, Farge-Bancel D, Fauvaux C, Feghoul L, Feredj E, Feyeux D, Fontaine JP, Fremeaux-Bacchi V, Galicier L, Garestier J, Jegu AL Kozakiewicz E Lebel M Baye A, Le Goff J, Le Guen P, Lengline E, Liegeon G, Lorillon G, Madelaine Chambrin I, Mahjoub N, Martin de Frémont G, Maylin S, Meunier M, Molina JM, Morin F, Oksenhendler E, Peffault de la Tour R, Peyrony O, Plaud B, Rouveau M, Salmona M, Saussereau J, Schnepf N, Soret J, Tazi A, Thegat M, Tremorin MT.

## Author Contributions

**Conceptualization:** Simon B. Gressens, Jean-Michel Molina.

**Data curation:** Simon B. Gressens, Violaine Esnault.

**Formal analysis:** Sylvie Chevret.

**Investigation:** Simon B. Gressens, Violaine Esnault, Nathalie De Castro, Pierre Sellier, Damien Sene, Louise Chantelot, Baptiste Hervier, Constance Delaugerre.

**Supervision:** Jean-Michel Molina.

**Visualization:** Sylvie Chevret.

**Writing – original draft:** Simon B. Gressens, Violaine Esnault.

**Writing – review & editing:** Nathalie De Castro, Pierre Sellier, Damien Sene, Louise Chantelot, Baptiste Hervier, Constance Delaugerre, Sylvie Chevret, Jean-Michel Molina.

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
