## [Decision Letter · Decision Letter 0]

2 Dec 2021

PONE-D-21-27196Remdesivir in combination with dexamethasone for patients hospitalized with COVID-19: a retrospective multicenter studyPLOS ONE

Dear Dr. Gressens,

Thank you for submitting your manuscript to PLOS ONE. After careful consideration, we feel that it has merit but does not fully meet PLOS ONE’s publication criteria as it currently stands. Therefore, we invite you to submit a revised version of the manuscript that addresses the points raised during the review process.

ACADEMIC EDITOR: Please review comments made by reviewers and provide your point by point response in your revised manuscript.

We look forward to receiving your revised manuscript.

Kind regards,

Muhammad Adrish, MD, MBA, FCCP, FCCM

Academic Editor

PLOS ONE

Journal Requirements:

"Jean-Michel Molina has participated to GILEAD Advisory Boards and has received research grants support for his institution in HIV research"

We note that you received funding from a commercial source: [Name of Company]

Reviewers' comments:

Reviewer's Responses to Questions

**Comments to the Author**

1. Is the manuscript technically sound, and do the data support the conclusions?

Reviewer #1: Yes

Reviewer #2: Yes

Reviewer #3: Yes

2. Has the statistical analysis been performed appropriately and rigorously? 

Reviewer #1: Yes

Reviewer #2: Yes

Reviewer #3: Yes

3. Have the authors made all data underlying the findings in their manuscript fully available?

Reviewer #1: No

Reviewer #2: No

Reviewer #3: Yes

4. Is the manuscript presented in an intelligible fashion and written in standard English?

Reviewer #1: Yes

Reviewer #2: Yes

Reviewer #3: Yes

5. Review Comments to the Author

Reviewer #1: Thanks for sending the paper for review. This is a well written draft. My specific comments are:

Line 42: Abstract: You can describe how you performed propensity score matching as you can add another 49 words.

Line 44-51: Methods: Please mention somewhere that you analyzed and presented data of 180 matched patients

Line 100: Informed consent can be taken from the relatives of the patients; I am not sure whether we could use data without any kind of consent. Please clarify this

Looking forward to a revised version.

Reviewer #2: Gressens et al report here the effect of the combination of dexamethasone + remdesivir compared to dexamethasone alone in hospitalized French patients with COVID-19 pneumonia. The authors state that in hospitalized patients with COVID-19 pneumonia receiving low-flow oxygen the addition of remdesivir to dexamethasone was not associated with shorter hospitalization or lower in-hospital mortality but may have reduced the combined outcome of death and transfer to the ICU.

The manuscript is enticing because it brings new light to the positive effect of remdesivir on COVID-19 infection. Results on this topic have been contradictory so far.

This reviewer can raise some points needing clarification by the authors.

Major points

1.Why different mortality outcomes were selected in this study: In-hospital death, in-hospital death and/or transfer to the ICU, in-hospital death and/or endotracheal intubation (Page 9, lines 134-140)

2.”In-hospital death rates were 8.9% and 20%, with and without remdesivir (p=0.06)!. This is a not-significant trend (p>0.05). perhaps due to the low number of patients included in the study (Page 16, line 275). This not-significant comparison might be confusing for the reader and perhaps it might be deleted from the abstract

Minor points

1. Could you actualize the worlwide COVID-19 information to November 2021? (Page 5, lines 59-60)

2. Could you explain further the expression “matched without replacement, using a caliper of 0.10 the standard deviation of the propensity…” (Page 19, lines 159-160). It is rather confusing in this form.

3. Could you change “endotracheal intubation (ETI)” to “mechanical ventilation”?

4. Could you change “OMS scale” to “WHO scale” in Table 1?

Reviewer #3: The manuscript by Gressens et al is concise and well-written. Although several retrospective studies of the same kind have been published this study adds another dataset to the our understanding of the potential benefits of remdesivir treatment.

Minor: The authors should mention the pinetree study in the discussion section.

6. PLOS authors have the option to publish the peer review history of their article (what does this mean?). If published, this will include your full peer review and any attached files.

Reviewer #1: **Yes: **Mahbub-Ul Alam

Reviewer #2: No

Reviewer #3: No

---

## [Author Response · Author response to Decision Letter 0]

5 Dec 2021

Reviewer #1: Thanks for sending the paper for review. This is a well written draft. My specific comments are:

Line 42: Abstract: You can describe how you performed propensity score matching as you can add another 49 words.

- Answer: We thank the reviewer for her/his kind comment. The description of the propensity score matching has been added in the abstract, as suggested by the Reviewer.

Line 44-51: Methods: Please mention somewhere that you analyzed and presented data of 180 matched patients

- Answer: This has been more clearly stated in the revised abstract and Methods sections of the manuscript.

Line 100: Informed consent can be taken from the relatives of the patients; I am not sure whether we could use data without any kind of consent. Please clarify this

Answer: In agreement with French bio-ethics law, as this was a retrospective study on data from patients charts, signed informed consent was not requested and patients or their relatives were informed that their anonymized data will be used for this analysis. They could refuse to have their data analyzed. This study was approved by the ethic research committee of the French ID society. This point has been clarified in the revised manuscript. 

Looking forward to a revised version.

Reviewer #2: Gressens et al report here the effect of the combination of dexamethasone + remdesivir compared to dexamethasone alone in hospitalized French patients with COVID-19 pneumonia. The authors state that in hospitalized patients with COVID-19 pneumonia receiving low-flow oxygen the addition of remdesivir to dexamethasone was not associated with shorter hospitalization or lower in-hospital mortality but may have reduced the combined outcome of death and transfer to the ICU.

The manuscript is enticing because it brings new light to the positive effect of remdesivir on COVID-19 infection. Results on this topic have been contradictory so far.

This reviewer can raise some points needing clarification by the authors.

Major points

1.Why different mortality outcomes were selected in this study: In-hospital death, in-hospital death and/or transfer to the ICU, in-hospital death and/or endotracheal intubation (Page 9, lines 134-140)

Answer: due to the small size of our cohort and the small number of deaths, we did not expect to see a difference in in-hospital mortality between arms. We wished however to assess the potential impact of adding remdesivir on severe outcomes, and to increase our statistical power we use composite outcomes including death and/or transfer to the ICU or death and/or mechanical ventilation. 

2.”In-hospital death rates were 8.9% and 20%, with and without remdesivir (p=0.06)!. This is a not-significant trend (p>0.05). perhaps due to the low number of patients included in the study (Page 16, line 275). This not-significant comparison might be confusing for the reader and perhaps it might be deleted from the abstract

Answer: We are not claiming that the difference between arms in in-hospital mortality is significant but feel that it is still important to provide data on mortality since it has been the main outcome used in clinical trials that have impacted guidelines (steroids, tocilizumab ..). 

Minor points

1. Could you actualize the worlwide COVID-19 information to November 2021? (Page 5, lines 59-60)

Answer: As suggested data have been uptdated in the revised manuscript. 

2. Could you explain further the expression “matched without replacement, using a caliper of 0.10 the standard deviation of the propensity…” (Page 19, lines 159-160). It is rather confusing in this form.

- Answer: This means that each patient in the Remdesivir group was matched to one patient in the dexamethasone group based on the closeness of their propensity score on a logit scale, using the nearest neighbor algorithm, with a tolerated maximal distance set at 0.10 fold the standard deviation of the PS. This has been rephrased in the revised manuscript.

3. Could you change “endotracheal intubation (ETI)” to “mechanical ventilation”?

Answer : it has been changed as suggested

4. Could you change “OMS scale” to “WHO scale” in Table 1?

Answer : it has been changed as suggested

Reviewer #3: The manuscript by Gressens et al is concise and well-written. Although several retrospective studies of the same kind have been published this study adds another dataset to the our understanding of the potential benefits of remdesivir treatment.

We thank the reviewer for this kind comment. 

Minor: The authors should mention the pinetree study in the discussion section.

Answer: The Pinetree study has not yet been presented when we submitted this manuscript but is now added in the discussion.

---

## [Decision Letter · Decision Letter 1]

30 Dec 2021

Remdesivir in combination with dexamethasone for patients hospitalized with COVID-19: a retrospective multicenter study

PONE-D-21-27196R1

Dear Dr. Gressens,

We’re pleased to inform you that your manuscript has been judged scientifically suitable for publication and will be formally accepted for publication once it meets all outstanding technical requirements.

Kind regards,

Maria Elena Flacco, M.D.

Academic Editor

PLOS ONE

Additional Editor Comments (optional):

Reviewers' comments:

Reviewer's Responses to Questions

**Comments to the Author**

1. If the authors have adequately addressed your comments raised in a previous round of review and you feel that this manuscript is now acceptable for publication, you may indicate that here to bypass the “Comments to the Author” section, enter your conflict of interest statement in the “Confidential to Editor” section, and submit your "Accept" recommendation.

Reviewer #2: All comments have been addressed

Reviewer #3: All comments have been addressed

2. Is the manuscript technically sound, and do the data support the conclusions?

Reviewer #2: Yes

Reviewer #3: Yes

3. Has the statistical analysis been performed appropriately and rigorously? 

Reviewer #2: Yes

Reviewer #3: Yes

4. Have the authors made all data underlying the findings in their manuscript fully available?

Reviewer #2: Yes

Reviewer #3: Yes

5. Is the manuscript presented in an intelligible fashion and written in standard English?

Reviewer #2: Yes

Reviewer #3: Yes

6. Review Comments to the Author

Reviewer #2: I am pleased with the revised version of this manuscript. The authors have answered all the queries raised by me and by the other reviewers . In my opinion this manuscript now it might be accepted in PLOS ONE

Reviewer #3: Comments have been adressed. No additional comments. The manuscript is well written and concise. The manuscript merits publication.

7. PLOS authors have the option to publish the peer review history of their article (what does this mean?). If published, this will include your full peer review and any attached files.

Reviewer #2: **Yes: **VICTOR ASENSI

Reviewer #3: No

---

## [Editor Report · Acceptance letter]

8 Feb 2022

PONE-D-21-27196R1 

Remdesivir in combination with dexamethasone for patients hospitalized with COVID-19: a retrospective multicenter study 

Dear Dr. Gressens:

I'm pleased to inform you that your manuscript has been deemed suitable for publication in PLOS ONE. Congratulations! Your manuscript is now with our production department. 

Kind regards, 

on behalf of

Dr. Maria Elena Flacco 

Academic Editor

PLOS ONE